# Resection of Meningiomas Invading the Cavernous Sinus: Treatment Strategy and Clinical Outcomes

**DOI:** 10.3390/cancers17020276

**Published:** 2025-01-16

**Authors:** Takashi Sugawara, Taketoshi Maehara

**Affiliations:** 1Department of Neurosurgery, International University of Health and Welfare, Chiba 2868520, Japan; 2Department of Neurosurgery, Institute of Science Tokyo Hospital, Tokyo 1130034, Japan; maehara.nsrg@tmd.ac.jp

**Keywords:** cavernous sinus, meningioma, clinical outcome, surgical technique, oculomotor

## Abstract

We describe a surgical strategy for the resection of meningiomas invading the cavernous sinus (CS) and the clinical outcomes, particularly regarding the restoration of cranial nerve function. We investigated 13 patients who had preoperative images of CS invasion, underwent surgical resection, and were followed-up with magnetic resonance imaging for at least 1 year. Preoperative symptoms, postoperative course, adjuvant therapy, postoperative complications, degree of resection, and recurrence were evaluated. In contrast to the rate of recovery of cranial nerve function by stereotactic radiosurgery (20–44.2%), oculomotor nerve paresis, abducens nerve paresis, and visual disturbance improved at least partially after our surgery in 87.5%, 83.3%, and 100% of patients, respectively. We believe that our study makes a significant contribution to the literature because we demonstrate that our surgical strategy for the resection of meningiomas in and around the CS to restore cranial nerve function is safe and effective, with only transient, acceptable injuries.

## 1. Introduction

Meningiomas in and around the cavernous sinus (CS) are likely to invade the sinus, optic sheath, and oculomotor cave. The cranial nerve and internal carotid artery are important structures in and around the CS [1,2]. Resection of a meningioma invading the CS carries the risk of damaging these important structures and may be fatal [2,3,4,5]. Therefore, radical surgery involving lesions in this anatomical region remains challenging, especially for meningiomas invading these structures [6,7,8].

Recently, meningiomas invading the CS have been treated with radiation therapies such as stereotactic radiosurgery (SRS) [9,10,11,12,13,14]. Tumors are well controlled with SRS, and the 10-year recurrent-free survival rate is reported to be 84.9–98%. However, the rate of recovery of cranial nerve function is only 20–44.2% [9,10,11,12,13]. Therefore, to increase the rate of cranial nerve function recovery, exposure and resection of these tumors is sometimes necessary and optimal for preventing recurrence [2] or for decompressing the cranial nerve to restore cranial nerve function [15]. However, this carries the risk of cranial nerve and internal carotid artery injury [2,3,4] and a high rate of aggressive surgery-related morbidity [16,17]; thus, resection of these tumors is generally avoided. Aggressive resection of the tumor in the CS is ideal, and resection to the maximum possible extent should be attempted. However, if the tumor in the CS is considered too stiff to resect without inflicting cranial nerve damage, wide openings of the optic nerve sheath and oculomotor cave are important for tumor resection and nerve decompression to restore optic and oculomotor nerve function.

In this study, we describe a surgical strategy and techniques for treating meningiomas invading the CS in order to restore cranial nerve function and report on the clinical outcomes of 13 patients.

## 2. Materials and Methods

### 2.1. Patients

The study was approved by the institutional review board of the Institute of Science Tokyo Hospital and the University of Health and Welfare. All patients in this case series provided informed consent for the inclusion of their clinical data in this manuscript.

This study investigated 13 patients with a meningioma, in whom preoperative magnetic resonance imaging (MRI) indicated tumor invasion of the CS, and who were followed-up with MRI for at least 1 year between July 2017 and July 2024. Preoperative symptoms, postoperative course, adjuvant therapy, postoperative complications, degree of resection, and recurrence or regrowth were evaluated.

### 2.2. Surgical Strategy

Surgical resection was indicated in patients with neurological symptoms or rapid tumor growth. The cranial nerves were preserved, and meningiomas invading the CS were excised as much as possible. Resection of meningiomas that invaded the CS was accomplished through the lateral, anterolateral, Dolenc’s, Parkinson’s, and paramedial triangles. The possibility of resecting such tumors was dependent on tumor stiffness and adhesion to the surrounding tissues. When a patient had neurological symptoms such as diplopia, ptosis, and visual field defects, the tumors were removed aggressively to restore nerve function without any further damage. Even if the tumor in the CS was considered too stiff to resect without inflicting cranial nerve damage, the optic nerve sheath and oculomotor cave were opened widely for tumor resection and nerve decompression, resulting in the restoration of optic and oculomotor nerve function.

When tumor control was poor even after multiple resections and radiotherapy and the clinical situation was considered to be critical, such as patients experiencing brainstem compression, or when the function of the cranial nerves was considered to be completely and irreversibly impaired, the tumor was removed by sacrificing the cranial nerves. When the internal carotid artery (ICA) was narrowed and tumor invasion into this wall was suspected, resection of the tumor around the ICA was avoided to prevent ICA rupture. During the resection of meningiomas that originated from or invaded the temporal dura propria, this structure was peeled away from the lateral wall of the CS and removed, followed by watertight closure of the pericranium.

### 2.3. Evaluation of Cranial Nerve Function

In terms of optic nerve function, if the visual field deficit was completely improved, it was classified as “completely improved”; partially improved but asymptomatic, as “almost completely improved”; partially improved and symptomatic, as “slightly improved”; and worsened, as “worsened.” For oculomotor and abducent nerve function, the score ranged from 0 (normal movement) to −5 (no movement), and if the score was completely improved to 0, it was classified as “completely improved”; almost increased to 0 and asymptomatic, as “almost completely improved”; improved to less than −1 and symptomatic, as “slightly improved”; and worsened, as “worsened”.

### 2.4. Adjuvant Irradiation

MRI was used to monitor patients with benign meningiomas (World Health Organization [WHO] grade I). When a tumor recurred or the residual tumor increased, the lesions were treated with intensity-modulated radiation therapy (IMRT) or SRS. For patients with malignant meningiomas (WHO grades II and III), IMRT was planned as soon as possible after tumor resection.

## 3. Results

### 3.1. Patient Characteristics

Clinical information of all patients is summarized in Table 1. Mean patient age was 59.1 years (range, 23–73 years), and 10 patients (76.9%) were female. Eight (61.5%) and five (38.5%) patients had meningiomas that were classified as WHO grades I and II, respectively. Eight (61.5%) patients had primary meningiomas, and five (38.5%) had recurrent meningiomas. The mean follow-up period was 37.4 months (17–76) (Table 2).

### 3.2. Preoperative Symptoms and Postoperative Course

Major preoperative symptoms included oculomotor nerve paresis in 8 patients (61.5%), abducens nerve paresis in 6 (46.2%), visual disturbance in 7 (53.8%), and brain swelling in 3 (23.1%) (Table 2). These symptoms improved at least partially after surgery in 7 (87.5%), 5 (83.3%), 7 (100%), and 3 (100%) patients, respectively. Oculomotor nerve paresis improved completely in 6 patients (75%), partially but almost completely in 1 (12.5%), and worsened in 1 (12.5%). Abducens nerve paresis improved completely in 3 patients (50%), partially but almost completely in 1 (16.7%), slightly in 1 (16.7%), and worsened in 1 (16.7%) (Table 3).

The patient that experienced worsening in all above-mentioned conditions had regrowth. The patient underwent partial resection of the tumor around the CS 12 years earlier; 7 years later, he underwent gamma knife surgery for regrowth in and around the CS. Five years after gamma knife surgery, tumor regrowth and oculomotor nerve and abducent nerve palsy worsened, and he underwent tumor resection to restore nerve function. However, the tumors around these nerves were too stiff to decompress, and these functions could not be restored. Failure to decompress these nerves and restore their function may therefore be attributed to the prior surgery and radiation therapy.

The only patient in whom abducens nerve palsy improved only slightly had a long history of symptoms and did not complain of diplopia, either preoperatively or postoperatively. Visual field disturbance improved completely in 3 patients (42.9%), partially but almost completely in 2 (28.6%), slightly in 2 (28.6%), and did not worsen in any (0%). All cases of brain swelling (100%) improved completely (Table 3).

### 3.3. Degree of Resection

Of the 13 meningiomas, 1 (7.7%) was removed by Simpson grade 1, 4 (30.8%) by Simpson grade 3, and 8 (61.5%) by Simpson grade 4 resections (Table 2). All residual tumors were located within or around the CS.

The tumors invaded the CS in all 13 patients (100%), the optic canal in 11 (84.6%), oculomotor cave in 10 (76.9%), orbit in 6 (46.2%), infratemporal fossa in 6 (46.2%), and the sinonasal cavity in 5 (38.5%). The CS was opened, and the tumor inside was resected slightly in 5 (38.5%), partially in 6 (46.2%), and completely in 2 (15.4%) patients. The windows that were opened through the CS were the anterolateral triangle (between the ophthalmic nerve [V1] and maxillary nerve [V2]) in 7 (53.8%), lateral triangle (between the V2 and maxillary nerve) in 6 (46.2%), Parkinson’s triangle (between the V1 and trochlear nerve) in 7 (53.8%), and paramedial triangle (between the oculomotor and trochlear nerves) in 2 patients (15.4%).

In all patients, the optic canal was opened with anterior clinoidectomy to create space around the optic and oculomotor nerves and to mobilize them, even in the absence of visual disturbance. The oculomotor cave was opened to remove the tumor and decompress the oculomotor nerve in 8 (61.5%) patients.

### 3.4. Postoperative Complications

Postoperative complications included temporary oculomotor paresis in 1 patient (7.7%), facial dysesthesia in 2 (15.4%), contralateral visual deterioration in 1 (7.7%), and brief transient slight hemiparesis caused by internal carotid vasospasm or dissection in 2 (15.4%) (Table 1). One patient with oculomotor paresis recovered 1 week after surgery. In patients with facial dysesthesia, symptom recovery was almost complete in 1 patient (7.7%), and V1 and V3 dysesthesia improved to a higher level than before surgery; however, V2 dysesthesia worsened after surgery in 1 patient (7.7%).

In one patient, contralateral visual deterioration was caused by endonasal endoscopic resection combined with craniotomy and an endonasal endoscopic approach for a recurrent atypical meningioma. In the patient with brief transient hemiparesis, the artery was completely exposed from the C1 to the C5 portion with the sacrifice of all cranial nerves in the CS. This was because the tumor was grade 2 and had not been controlled by numerous previous surgeries and radiotherapy. The patient’s visual acuity was completely lost, and this was irreversible. Therefore, the cause of this brief transient hemiparesis was thought to be internal carotid artery vasospasm caused by mechanical stimulation.

### 3.5. Adjuvant Therapy and Postoperative Course

The mean duration of postoperative follow-up was 34.7 months (range, 7–76 months). Four patients with residual atypical meningioma of the CS underwent IMRT. Six lesions recurred or regrew, resulting in additional treatment with SRS in 2 patients, IMRT in 3, and resection in 1. SRS was performed for 2 recurrent atypical meningiomas that had already undergone IMRT after resection, and IMRT was performed for 2 grade 1 recurrent meningiomas. One patient with recurrent clear cell meningioma underwent resection 27 months after the second surgery, but the pathology was changed to anaplastic meningioma (WHO grade 3), and the patient died due to rapid growth of the meningioma 4 months post-surgery.

### 3.6. Illustrative Patient Cases

Patient #9 was a 47-year-old woman with a cavernous meningioma invading the optic canal and oculomotor cave (Figure 1A,B). She had progressive left-sided oculomotor and abducens nerve paresis (Figure 2A–E) and mild facial dysesthesia. Most of the tumor was resected; however, a small amount remained in the CS and around the trochlear nerve (Figure 1C,D). The tumor in the CS was resected through the anterolateral (Figure 1E), lateral, paramedial (Figure 1F), and Parkinson’s triangles (Figure 1G). The abducens nerve was identified through the anterolateral (Figure 1E) and Parkinson’s triangles (Figure 1G). Her diplopia and ptosis (Figure 2A–E) improved immediately after surgery and completely resolved within 1 week (Figure 2F–J). She had no additional postoperative symptoms.

Patient #7 was a 66-year-old man with a craniofacial atypical meningioma invading the CS, optic canal, orbit, and the sphenoid and ethmoid sinuses (Figure 3A,B). He presented with severe left-sided oculomotor and abducens nerve palsy, diplopia, ptosis, and exophthalmos. The tumor outside the CS was completely resected, and most of the tumor present inside the CS was removed as a Simpson grade 3 resection through a combination of a transcranial approach and an endoscopic endonasal approach (Figure 3C,D). The portion of the tumor invading the CS was partially resected, enough to decompress the cranial nerves through the anterolateral (Figure 3E), lateral (Figure 3F), and Parkinson’s triangles (Figure 3G). Severe diplopia, ptosis, and exophthalmos completely resolved within 4 weeks, and the patient had no additional postoperative symptoms. The patient underwent intensity-modulated radiotherapy 2 months post-surgery.

Patient #4 was a 74-year-old woman with a cavernous meningioma extending to the middle cranial fossa, optic canal, oculomotor cave, and sellar region (Figure 4A,B). She presented with severe oculomotor nerve palsy, optic nerve palsy, left-sided exophthalmos, severe diplopia, and ptosis. The Parkinson’s triangle was exposed; however, only a small amount of the tumor in the CS was removed because it was expected to be too tough and difficult to resect without causing cranial nerve damage. Part of the tumor invading the optic canal and oculomotor cave (Figure 4E) was completely removed with wide exposure of these structures (Figure 4F–H). The oculomotor nerve was compressed and displaced medially and was encased by the tumor in the oculomotor cistern (Figure 4E). The anterior petroclinoid fold was cut, and the oculomotor cave was widely exposed (Figure 4F) to remove the tumor and decompress the nerve. The oculomotor nerve in the cistern (# in Figure 4) and in the oculomotor cave (## in Figure 4) were completely exposed and decompressed. Severe diplopia and ptosis completely resolved within 8 weeks, visual field deficits partially improved, and she had no additional postoperative symptoms.

Patient #1 was a 23-year-old woman with a recurrent atypical craniofacial meningioma invading the bilateral CS, optic canal, orbit, sphenoid and ethmoid sinuses, posterior fossa, and infratemporal fossa (Figure 5A–C). She had undergone three surgeries: external beam radiotherapy and 4 gamma knife procedures before this recurrence. She exhibited complete left external ophthalmoplegia, left eye blindness, and facial dysesthesia. The tumors in the right and posterior part of the left CS did not grow, but the tumors in the anterior part of the left CS, posterior fossa, and infratemporal fossa had continued to grow since the most recent gamma knife procedure. These growing lesions were resected by sacrificing all the cranial nerves in the CS using a combination of transcranial and endoscopic endonasal approaches (Figure 5F,G), and only the internal carotid artery was preserved. All growing components were removed, and the brain stem was decompressed (Figure 5D,E). Vision in the left eye improved slightly (to light perception); however, vision in the right eye was lost. Three days after surgery, she experienced transient slight right hemiparesis with internal carotid artery vasospasm 3 days post-surgery. There was a small recurrence in the posterior fossa; therefore, she underwent gamma knife surgery 10 months after the resection.

## 4. Discussion

Preoperative cranial nerve deficits occur in 70–89% of patients with CS meningioma, and in 14–66% of these cases, neuropathy improves postoperatively [2,18,19,20,21]. In contrast to the rate of recovery of cranial nerve function by SRS (20–44.2%) [9,10,11,12,13], oculomotor nerve paresis, abducens nerve paresis, and visual disturbance improved at least partially after our surgery in 87.5%, 83.3%, and 100% of our patients, respectively. The sample size of 13 patients is too small to draw definitive conclusions, but the superiority of the cranial nerve recovery rate, obtained after our surgical strategy, compared to the reported outcomes of SRS, may demonstrate the efficacy of this approach.

### 4.1. Postoperative Course of Preoperative External Ophthalmoparesis and Visual Disturbances

Ophthalmoparesis abated without any interference with daily life in 8 of the 9 (89%) affected patients, and visual disturbances were observed in 7 of the 7 (100%) patients. The only patient in which ophthalmoparesis persisted was the patient who had the meningioma regrowth after previous resection and SRS. The tumors around the oculomotor and abducens nerves were too stiff to decompress, and these functions could not be restored. Thus, failure to decompress these nerves and restore their function can be attributed to the prior surgery and radiotherapy. This suggests that radiotherapy such as SRS may make it difficult to restore cranial nerve function and control tumors by surgical resection after recurrence by making the tumor stiff and tightly adherent [22]. This also indicates that even after recurrence, it may be beneficial to consider tumor removal, if it is resectable, in order to restore cranial nerve function. In Patient #10 in our study, who underwent IMRT after surgery, the tumor recurred, progressing to WHO grade III, and the patient died 14 years after the initial surgery. This observation suggests that radiotherapy may affect cranial nerve recovery and tumor control by leading progression into a malignant tumor. It is reported that tumor progression (treatment failure) after SRS may demonstrate a transformation, and careful, close, and long follow-up is highly recommended [23]. However, some systematic reviews showed that radiosurgery did not appear to increase the incidence rate of malignant transformation; nevertheless, exact comparisons are difficult because of differences in study populations [23,24,25,26]. Therefore, the effect of radiosurgery on malignant transformation remains controversial.

Decompression of the oculomotor nerve by wide oculomotor cave exposure and tumor removal is effective for recovering oculomotor function. Hegazy et al. reported that routine mobilization of the outer CS membrane reduces bleeding, which helps surgeons better visualize the cranial nerves and enables more radical resection of large sphenoclinoidal meningiomas [27]. Our surgical strategy of peeling off and removing the temporal dura propria may have contributed to good cranial nerve function outcomes.

As demonstrated by Patient #4, even if the tumor in the CS is too stiff to be removed and it is considered better to leave the tumor in place, oculomotor function can be effectively restored by widely opening the oculomotor cave to resect the tumor inside. The recovery rates of oculomotor and abducent nerve function with decompressive surgery for cavernous sinus meningioma are reported to be 50% and 0%, respectively [18]. Our surgical strategy can alleviate external ophthalmoparesis and visual disturbances by aggressive removal of the tumor from the CS, optic canal, and oculomotor caves. This implies that our strategy for meningiomas invading the CS is effective and reasonable. However, this comparison of recovery rates of nerve function has only been performed with previously reported studies of SRS, which weakens our conclusion regarding the superiority of surgery. Thus, a randomized control trial is needed in the future.

### 4.2. Postoperative Complications

One patient developed new-onset oculomotor paresis postoperatively; however, this symptom resolved completely within 1 week. Basma et al. reported that mobilization of the transcavernous oculomotor nerve resulted in better maneuverability and reduced nerve tension, leading to a favorable oculomotor outcome [28]. The oculomotor nerve can be easily damaged by peeling from tumors without proper mobilization of the cavernous oculomotor nerve. However, it is also likely to recover completely with proper mobilization by oculomotor cave exposure. Two patients developed postoperative facial dysesthesia and recovered with mild persistent symptoms that could be ignored.

In a patient who experienced hemiparesis with internal carotid artery vasospasm 3 days post-surgery, recovery was complete within a few hours. During her surgery, the internal carotid artery from the C1 to C5 portion was completely exposed, and all cranial nerves in the CS were sacrificed. Bejjani et al. reported that 1.9% of patients with cranial base tumors had postoperative cerebral vasospasms, and 89% of these patients exhibited symptoms [29]. They concluded that preoperative vessel encasement and narrowing were correlated with a higher incidence of postoperative vasospasm. Therefore, it is important to monitor for vasospasms during the resection of tumors in and around the CS. In our strategy, when the ICA is narrow and tumor invasion into the ICA wall is suspected, resection of the tumor around the ICA is avoided to prevent ICA rupture. In another patient with transient hemiparesis, the C1 portion was dissected and narrowed; however, the symptoms were mild and transient. In this case, the C1-2 portion of the ICA was also surrounded by a meningioma in the ICA cistern, and this ICA dissection and occlusion could be attributed to the resection of the meningioma from the C1-2 portion.

The complications experienced by our patients were transient or minimally disabling and considered acceptable. The accurate evaluation of meningiomas in and around the CS, as well as the careful and delicate lesion removal techniques performed during our surgical procedure, can lead to a high rate of cranial nerve function restoration. Furthermore, we believe that complete resection of the meningioma in the CS can be achieved by advancing the techniques and strategies used in our study in the near future.

### 4.3. Limitations

The limitations of this study include its small sample size, retrospective design, the short mean follow-up period of 37.4 months, and the inclusion of different tumor grades (WHO grade I and II) and original locations. While we were unable to determine late recurrences, the restoration rate of cranial nerve function was found to be high in this study, even with a short follow-up period, different tumor grades, and original locations. Accordingly, we intend to accumulate more cases and follow-up with patients for a longer period. Further, large-scale studies are needed to validate our outcomes. Finally, this surgery for CS meningiomas is relatively rare, and it may be difficult to accumulate cases to draw conclusions. In such cases, the use of synthetic data could address the issue of data scarcity [30,31].

## 5. Conclusions

In order to restore cranial nerve function, our surgical strategy for the resection of meningiomas in and around the CS is safe and effective, with only transient, acceptable injuries. Even if the tumor in the CS is too stiff to be removed, the optic nerve sheath and oculomotor cave can be opened widely to effectively remove the tumor.

## Figures and Tables

**Figure 1 cancers-17-00276-f001:**
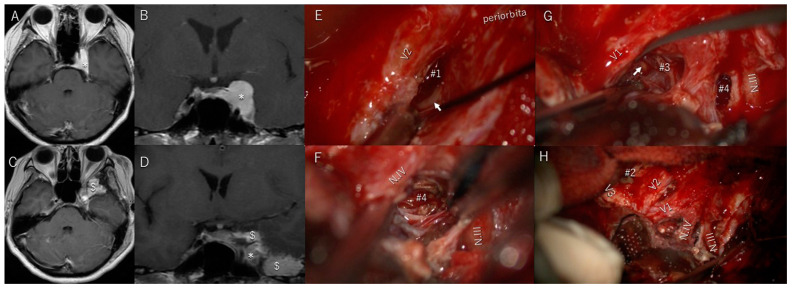
Patient #9 (47-year-old woman): cavernous meningioma (CS). (**A**,**B**) Preoperative T1-weighted magnetic resonance imaging (MRI) with contrast media showing the tumor (*) in and around the CS. (**C**,**D**) Postoperative T1-weighted MRI with contrast showing that only a small amount of tumor remained in the CS. After resection, the cavity is filled with fat tissue ($). (**E**–**G**) The anterolateral (#1), paramedial (#4), and Parkinson’s triangles (#3) are widely exposed to remove the tumor in the CS, and the abducens nerve (arrow) is identified through the anterolateral (**E**) and Parkinson’s triangles (**G**). (**H**) Lateral wall of CS after tumor resection. Abbreviations: V1, V2, and V3, first, second, and third divisions of the fifth cranial nerve; *, tumor; $, bedded fat tissue; #1, anterolateral triangle; #2, lateral triangle; #3, Parkinson’s triangle; arrow, meningohypophyseal trunk; N.IV, trochlear nerve. (**G**) The Parkinson’s triangle (#3) is exposed to remove the tumor from the CS and Meckel’s cave. Arrow indicates the meningohypophyseal trunk. Abbreviations: N.III, oculomotor nerve; N.IV, trochlear nerve; V1, first division of the fifth cranial nerve.

**Figure 2 cancers-17-00276-f002:**
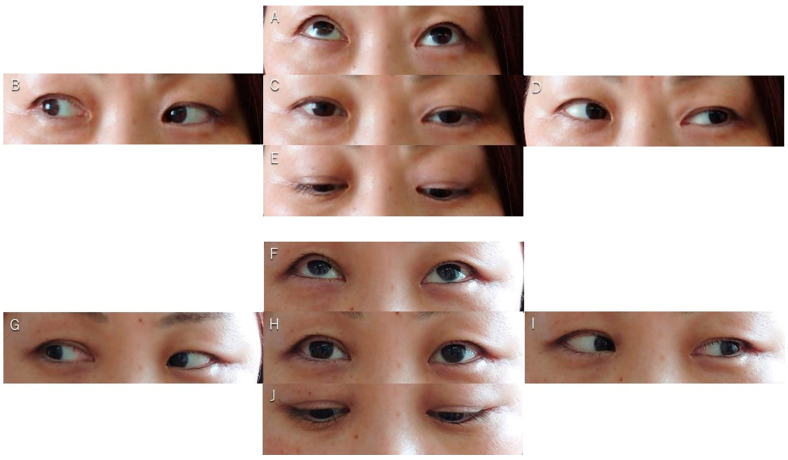
Preoperative (**A**–**E**) and postoperative (**F**–**J**) eye movement.

**Figure 3 cancers-17-00276-f003:**
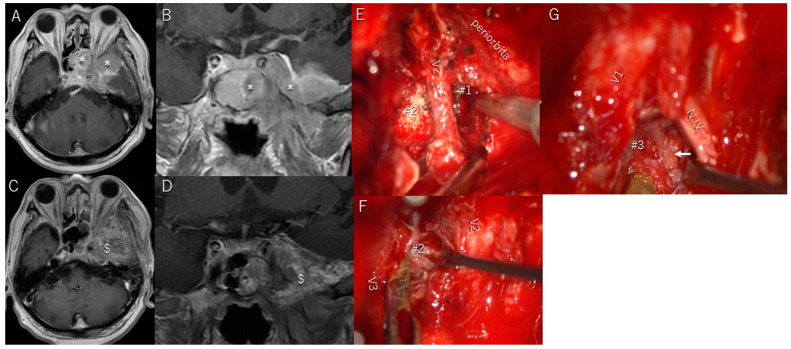
Patient #7 (66-year-old man): craniofacial atypical meningioma. (**A**,**B**) Preoperative T1-weighted magnetic resonance imaging (MRI) with contrast media showing tumor (*) invasion of the cavernous sinus (CS), optic canal, orbit, and sphenoid and ethmoid sinuses. (**C**,**D**) Postoperative T1-weighted MRI with contrast media showing that all tumors except one in the CS were resected; the tumor (*) in the CS was partially resected. After resection, the cavity is filled with fat tissue ($). (**E**,**F**) The anterolateral (#1) and lateral triangles (#2) are widely exposed to remove the tumor in the CS. Abbreviation: V2 and V3, second and third divisions of the fifth cranial nerve. (**G**) Parkinson’s triangle (#3) is exposed to remove the tumor in the CS and Meckel’s cave. The arrow points to the meningohypophyseal trunk. Abbreviations: N.IV, trochlear nerve; V1, first division of the fifth cranial nerve.

**Figure 4 cancers-17-00276-f004:**
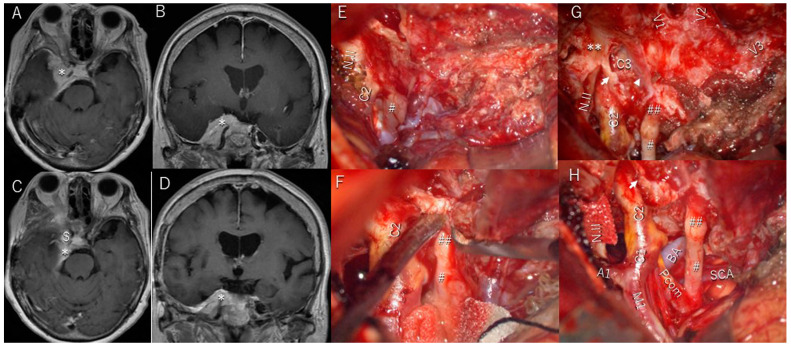
Patient #4 (74-year-old woman): cavernous meningioma (CS). (**A**,**B**) Preoperative T1-weighted magnetic resonance imaging (MRI) with contrast media showing that the tumor (*) extends to the middle cranial fossa, optic canal, oculomotor cave, and sellar region. (**C**,**D**) Postoperative T1-weighted MRI with contrast media showing that the only remaining tumor (*) is mainly in the CS. The cavity formed after tumor resection is filled with fat tissue ($). (**E**) Tumor invading the optic canal and oculomotor cave, and the oculomotor nerve (#) is displaced medially and encased by the tumor in the oculomotor cave. Abbreviations: C2, internal carotid artery, C2 segment; N.II, optic nerve. (**F**) The anterior petroclinoid fold is cut along in the direction of the oculomotor nerve axis, and the oculomotor cave is widely exposed. Abbreviations: C2, internal carotid artery, C2 segment. (**G**,**H**) The tumor (*) in the optic canal and oculomotor cave is completely removed through wide exposure of these structures. The oculomotor nerve in the cistern (#) and oculomotor cave (##) are completely exposed and decompressed. The optic sheath is denoted by **; the arrow points to the distal dural ring; the arrowhead points to the proximal dural ring. Abbreviations: A1, anterior cerebral artery; A1 segment; BA, basilar artery; C1, C2 and C3, internal carotid artery; C1, C2 and C3 segments; M1, middle cerebral artery; M1 segment; N.II, optic nerve; Pcom, posterior communicating artery; SCA, superior cerebral artery; V1, V2, and V3, divisions of the fifth cranial nerve.

**Figure 5 cancers-17-00276-f005:**
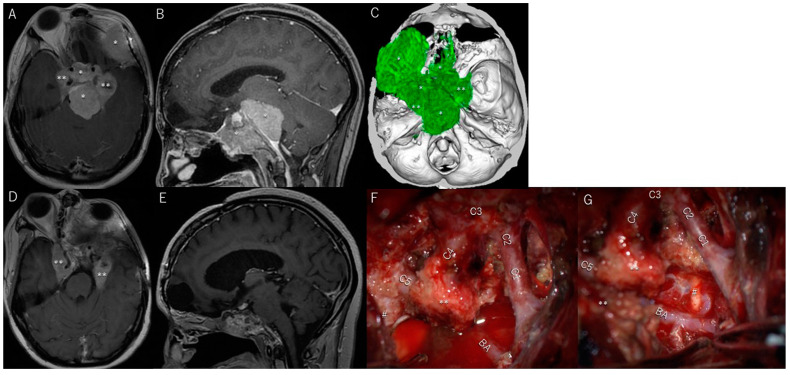
Patient #1 (23-year-old woman): recurrent craniofacial atypical meningioma. (**A**,**B**) Preoperative T1-weighted magnetic resonance imaging (MRI) with contrast media showing tumor (*) invasion of the bilateral cavernous sinus (CS), optic canal, orbit, sphenoid and ethmoid sinuses, posterior fossa, and infratemporal fossa. The fibrotic granuloma stabilized by previous resection and radiosurgery is indicated by **. (**C**) Three-dimensional computed tomography showing tumor spread (green area). (**D**,**E**) Postoperative T1-weighted MRI with contrast media revealing that the entire growing tumor was removed except the fibrotic granuloma (**), and that the brain stem is decompressed. (**F**) The growing tumor (*) is resected, all of the cranial nerve in the CS is sacrificed, and the five segments of the internal carotid artery (C1 to C5) are exposed. Abbreviations: BA, basilar artery. (**G**) The tumor expanding into the posterior fossa is resected, the brain stem is decompressed, and the basilar artery (BA) and contralateral oculomotor nerve are (#) visualized. * tumor, ** fibrotic granuloma stabilized by previous radiosurgery. Abbreviations: C1 to C5, internal carotid artery, C1 to C5 segments.

**Table 1 cancers-17-00276-t001:** Summary of all 13 patients.

Patient No.	Age (y/o) /Sex	Primary /Recurrent	WHO Grade	Invasion						
				Optic Canal	Oculomotor Cave	Cavernous Sinus	Orbit	Periorbita	Infratemporal Fossa	Sinonasal Sinus
1	23/F	r	2	y	y	y	y	n	y	y
2	44/F	p	1	y	y	y	n	n	n	n
3	69/F	p	1	y	y	y	n	n	n	n
4	74/F	p	1	y	y	y	y	y	y	y
5	65/F	p	1	n	n	y	y	n	y	y
6	58/F	p	2	y	y	y	n	n	n	n
7	66/M	r	2	y	n	y	y	n	y	y
8	65/F	r	2	y	n	y	y	y	y	y
9	47/F	p	1	y	y	y	n	n	n	n
10	56/F	r	2	y	y	y	n	n	n	n
11	67/F	p	1	n	y	y	n	n	n	n
12	61/M	r	1	y	y	y	y	n	y	n
13	73/M	p	1	y	y	y	n	n	n	n
mean	59.1									
**Patient No.**	**Open**			**Resection of the Tumor in Cavernous Sinus**		**Simpson Grade**	**Preoperative Symptom/Postoperative Outcome**			
	**Optic** **Canal**	**Oculomotor Cave**	**Cavernous Sinus**				**Optic Nerve**	**Oculomotor Nerve**	**Abducent Nerve**	**Other**
1	y	y	entire lateral wall of cavernous sinus	Totally with sacrifice of all cranial nerves		4	y/no change	y/sacrifice	y/sacrifice	brain stem swelling/complete recovery
2	y	y	lateral triangle	Slightly		4	y/almost complete recovery	y/complete recovery	y/complete recovery	
3	y	y	lateral triangle	Slightly		3	y/almost complete recovery	n	n	
4	y	y	Parkinson triangle	Slightly		3	y/slightly improved	y/complete recovery	n	
5	y	n	anterolateral triangle	Totally		1	n	n	n	brain swelling/complete recovery
6	y	n	Parkinson triangle	Slightly		4	y/complete recovery	n	n	brain swelling/complete recovery
7	y	n	Parkinson, anterolateral, lateral triangle	Partially		3	y/complete recovery	y/complete recovery	y/complete recovery	exophthalmos/complete recovery
8	y	n	Parkinson, anterolateral, lateral triangle	Partially		3	y/slightly improved	y/almost complete recovery	y/almost complete recovery	facial dysesthesia/almost complete recovery
9	y	y	anterolateral, paramedial, lateral, Parkinson triangle	Subtotally		4	n	y/complete recovery	y/complete recovery	facial dysesthesia/almost complete recovery
10	y	n	Parkinson triangle	Partially		4	n	y/worsened.	y/worsened.	
11	y	y	paramedial, Parkinson, lateral, anterolateral triangle	Subtotally		4	y/complete recovery	n	y/slightly improved	
12	y	y	anterolateral triangle,	Partially		4	n	y/complete recovery	n	exophthalmos/complete recovery
13	y	y	anterolateral triangle	Partially		4	n	y/complete recovery	n	
mean										
**Patient No.**	**Complication**				**Adjuvant Therapy**		**Recurrence or Regrowth**	**Follow-Up Period (Months)**	**Additional Treatment**	
1	Contralateral visual deterioration Brief transient slight hemiparesis (vasospasm s/o)				n		y	41	SRS	
2	n				n		n	76		
3	n				n		y	73	SRS 25 Gy/5 fr	
4	n				n		y	66	IMRT 60 Gy/30 fr	
5	n				n		y	11	resection + IMRT 60 Gy/30 fr	
6	brief transient slight hemiparesis (C2 segment dissection s/o)				IMRT 60 Gy/30 fr		n	57		
7	n				IMRT 60 Gy/30 fr		n	23		
8	slightly transient facial dysesthesia				IMRT 50.4 Gy/28 fr		y	17	SRS	
9	n				n		n	26		
10	n				IMRT(60 Gy/30 fr)		y	27	resection (malignant change to anaplastic meningioma)	
11	Abducens nerve palsy slightly transiently worsened				n		n	23		
12	slightly transient facial dysesthesia				n		n	24		
13	n				n		n	22		
mean								37		

**Table 2 cancers-17-00276-t002:** Clinical characteristics and patient outcomes.

Mean Age, Years (Range)	59.1 (23–73)
Sex	
Male	3 (23.1)
Female	10 (76.9)
WHO grade	
Grade 1	8 (61.5)
Grade 2	5 (38.5)
Primary/recurrent	
Primary	8 (61.5)
Recurrent	5 (38.5)
Preoperative symptoms	
Oculomotor nerve paresis	8 (61.5)
Abducens nerve paresis	6 (46.2)
Visual disturbance	7 (53.8)
Brain swelling	3 (23.1)
Simpson grade	
Total removal (grade 1–2)	1 (7.7)
Partial removal (grade 3–4)	12 (92.3)
Complication	
Temporary oculomotor paresis	1 (7.7)
Facial dysesthesia	2 (15.4)
Contralateral visual deterioration	1 (7.7)
Brief transient slight hemiparesis	2 (15.4)
Mean follow-up period, months (range)	37.4 (17–76)

**Table 3 cancers-17-00276-t003:** Clinical outcome of preoperative symptoms.

	Completely Improved	Almost Completely Improved	Slightly Improved	Worsened
Oculomotor nerve paresis (n = 8)	6 (75)	1 (12.5)	0 (0)	1 (12.5)
Abducens nerve paresis (n = 6)	3 (50)	1 (16.7)	1 (16.7)	1 (16.7)
Visual disturbance (n = 7)	3 (42.9)	2 (28.6)	2 (28.6)	0 (0)
Brain swelling (n = 3)	3 (100)	0 (0)	0 (0)	0 (0)

## Data Availability

The datasets analyzed in the current study are available from the corresponding author upon reasonable request.

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
