# Peer review of "Resection of Meningiomas Invading the Cavernous Sinus: Treatment Strategy and Clinical Outcomes"

_cancers, 2025, doi:10.3390/cancers17020276_

Round 1

Reviewer 1 Report

Comments and Suggestions for Authors

Reviewer’s comments

This manuscript, "Resection of Meningiomas Invading the Cavernous Sinus: Treatment Strategy and Clinical Outcomes Summary," is well-written and has great potential. However, some areas require revision to improve clarity and strengthen the manuscript's scientific robustness.

Here are my comments to improve the quality of the manuscript

Minor Corrections

1.       While the objectives are clear, the introduction lacks sufficient context on how this study differentiates from prior works. Including a more specific discussion on gaps in existing literature and how the proposed surgical strategy addresses them would strengthen the manuscript.

2.       The sample size of 13 patients is small, which limits the generalizability of the findings. Address this limitation more explicitly in the discussion.

3.       Consider elaborating on the implications for future research and how findings could influence clinical guidelines.

4.       The reference list is comprehensive but does not cite recent literature from 2022–2023 on surgical or radiological advancements in meningioma management. To enhance the context and citation depth in discussions about surgical techniques and predictor modeling, the following papers could be considered for inclusion: Enhanced MRI-based brain tumor classification with a novel Pix2Pix generative adversarial network augmentation framework.” And “A Comparative Analysis of the Novel Conditional Deep Convolutional Neural Network Model, Using Conditional Deep Convolutional Generative Adversarial Network-Generated Synthetic and Augmented Brain Tumor Datasets for Image Classification”.

Reviewer 2 Report

Comments and Suggestions for Authors

The manuscript titled "Resection of Meningiomas Invading the Cavernous Sinus: Treatment Strategy and Clinical Outcomes" addresses the surgical management of meningiomas invading the cavernous sinus (CS). It aims to evaluate a surgical strategy for tumor resection and its effectiveness in restoring cranial nerve function. The study investigates clinical outcomes in 13 patients who underwent surgery for CS-invading meningiomas, focusing on cranial nerve recovery, postoperative complications, tumor resection degree, and recurrence. The key findings are I.) study reports significant improvement in cranial nerve function, particularly in oculomotor nerve paresis (87.5%), abducens nerve paresis (83.3%), and visual disturbance (100%). Complete tumor resection within the CS was rare due to high risks, with most resections achieving partial removal (Simpson grade 3 or 4). Postoperative complications were minimal and transient in most cases, including temporary oculomotor paresis and hemiparesis. The surgical approach demonstrated higher rates of nerve recovery than stereotactic radiosurgery (SRS).

There are major limitaiosn: the study is retrospective, has a small sample size, and a relatively short follow-up period (mean 37.4 months). The small sample size (13 patients) limits the generalizability of the findings. Larger studies are needed to validate the reported outcomes. Patients with severe symptoms or recurrent tumors might have been prioritized for surgery, leading to selection bias. This limits the applicability of results to all meningioma cases. The study combines different tumor grades (WHO grades I–III) and locations, which might confound the outcomes since higher-grade tumors have distinct biological behaviors. Improvement in cranial nerve function is partially subjective and may vary based on the patient's or surgeon's assessment. More standardized quantitative evaluations could provide greater objectivity. While the study highlights the challenges of achieving total resection in the CS, it does not thoroughly discuss the implications of residual tumors on long-term outcomes, especially for high-grade meningiomas. Some patients received intensity-modulated radiotherapy (IMRT) or SRS post-surgery, which could influence cranial nerve recovery and tumor control. The interplay between surgery and adjuvant treatments needs further exploration. Recurrences were noted in several cases, and the follow-up period might be insufficient to capture late recurrences, especially for benign meningiomas. While the study mentions higher nerve recovery rates compared to SRS, direct comparisons with a control group undergoing SRS alone are lacking. This diminishes the strength of the conclusion about the superiority of surgery. Addressing these design issues and expanding the study would enhance its impact and reliability in guiding clinical practice.

Round 2

Reviewer 2 Report

Comments and Suggestions for Authors

I am afraid, there are still quite a few major problems with the study. The small sample size, with only 13 cases included, the study has limited statistical power. This makes it difficult to generalize findings or draw strong conclusions, as noted in the manuscript itself. The inclusion of patients with different tumor grades (WHO I and II), types (primary vs. recurrent), and anatomical involvements introduces variability that could affect outcomes. This heterogeneity challenges the applicability of the findings to a broader patient population. The study does not compare outcomes with alternative treatments such as stereotactic radiosurgery or conservative management, limiting the ability to assess the superiority of the proposed surgical strategy. The mean follow-up of 37.4 months may not be sufficient to detect late recurrences or long-term complications, especially for recurrent or higher-grade meningiomas. The manuscript emphasizes favorable outcomes, but the subjective evaluation of nerve function improvement could introduce bias, particularly without standardized or blinded assessments. Although complications are briefly mentioned, a more thorough discussion of the risks, such as potential for cranial nerve damage and long-term functional impairments, is warranted. The findings are based on a single-center experience with a specific surgical technique. Replicability of results in other settings or by other surgeons is not addressed. The use of various adjuvant therapies (e.g., IMRT, SRS) complicates the interpretation of surgical outcomes, as these treatments could independently influence tumor control and nerve function recovery.

However, the management tactics and microsurgical practice of the authors may be interesting to neurosurgeons reading Cancers.